# A large-scale trial in Chennai, India shows effective reduction of *Anopheles stephensi* populations by the biolarvicide Natular

Candasamy Sadanandane[1], Ananganallur Nagarajan Shriram[1]*, Ashwani Kumar[1‡], Mariapillai Kalimuthu[2], Ranganathan Krishnamoorthi[1], S. Selvakumar[3], Annamalai Sakthivel[1], Manju Rahi[1]

**1** Department of Health Research, ICMR-Vector Control Research Centre, Indian Council of Medical Research, Ministry of Health and Family Welfare, Government of India, Puducherry, India, **2** Department of Health Research, ICMR-Vector Control Research Centre, Indian Council of Medical Research, Ministry of Health and Family Welfare, Government of India, Madurai, Tamil Nadu, India, **3** Greater Chennai Corporation, Government of Tamil Nadu, Chennai, Tamil Nadu, India

‡ Former Director, ICMR-VCRC.
* anshriram@gmail.com, shriram.an@icmr.gov.in

## Abstract

Biolarvicides present an environmentally safer alternative to chemical larvicides, demonstrating selectivity against target species. However, the available options for biolarvicide use in mosquito control remain limited. This study aimed to assess the effectiveness of three formulations of the biolarvicide Natular, namely the 20.6% Emulsifiable Concentrate (EC), 2.5% G30 granular, and 7.48% tablet (DT) for direct application, against *Anopheles stephensi* in large-scale field trials conducted in Chennai, India. When the 7.48% DT formulation was applied at a rate of 1 tablet per 200 L in overhead tanks with *An. stephensi* breeding sites, on an average, led to a > 90–100% reduction in *An. stephensi* larval and pupal density for a duration of 8–9 weeks in rainy, winter and summer seasons. Similarly, the 2.5% G30 formulation, at a concentration of 0.074 mg (AI)/L in overhead tanks, impressively suppressed *An. stephensi* larval and pupal density by 91–100% for 4–5 weeks in three seasons. The application of the 20.6% EC formulation at 0.073 mg (AI)/L in overhead tanks achieved a remarkable 100% reduction in larval and pupal density for 2 weeks in all seasons. The Natular 7.48% DT formulation is well-suited for *An. stephensi* larval control at 1 tablet per 200 L with an 8-week application interval. The 2.5% G30 formulation is recommended at 0.074 mg (AI)/L at 4-week intervals. Meanwhile, the 20.6% EC formulation can effectively control *An. stephensi* at 0.073 mg (AI)/L with fortnightly application intervals. These Natular formulations serve as potent biolarvicides, offering a valuable tool for *An. stephensi* control within the Integrated Vector Management (IVM) program.

**Data availability statement:** Relevant summary data are available within the paper in the form of primary tables and graphs.

**Funding:** This study was supported by funding from Clarke International, USA. (https://www.clarke.com/international). The funders had no role in the study design, data collection and analysis, decision to publish, or preparation of the manuscript.

**Competing interests:** The authors have declared that no competing interests exist.

## Introduction

Larval Source Management (LSM) is effective in certain eco-epidemiological situations because it targets both resting and biting mosquitoes, both indoors and outdoors, which is a crucial aspect of reducing residual transmission [1]. As a result, it becomes an essential element in the overall strategy for integrated vector management (IVM) [2]. LSM has shown that larval control methods, specifically larviciding, can reduce the number of adult mosquitoes and eliminate mosquito larval breeding [3,4].

Despite the development and evaluation of numerous chemical larvicidal formulations that are effective against mosquito vector larval stages, financial constraints, the emergence of vector resistance, and environmental considerations hinder their application. Temephos, a key organophosphate insecticide is utilized to control mosquitoes that transmit malaria and arboviruses. However, it has been having trouble because a lot of vector mosquitoes have become resistant to it, especially in places where it has been used for a long time [5–7].

In these situations, biolarvicides, especially microbial agents, are better suited than chemical larvicides because they are safer for the environment and target only the species they are intended to kill. Still, there aren't many microbial larvicides available to kill mosquito larvae. The available ones are primarily based on *Bacillus thuringiensis israelensis* (Bti) and *Bacillus sphaericus* (Bs). Furthermore, previous studies have documented the emergence of resistance in mosquito vectors, specifically against *B. sphaericus* [8–10]. Hence, we must accelerate the development of novel biolarvicides that possess unique modes of action and advantageous toxicological and environmental safety profiles to augment their effectiveness in an integrated vector management strategy.

Recent years have seen the emergence of Spinosad as a promising biorational insecticide, offering an alternative to environmentally hazardous and resistant synthetic insecticides. Spinosad is derived from the naturally occurring soil bacterium, *Saccharopolyspora spinosa*. It is not very toxic to mammals and is safe for the environment. Different forms of Spinosad, including 0.5% granular (GR), 12% suspension concentrate (SC), 20% emulsifiable concentrate (EC), and 8.33% tablet (DT) for direct application, have been evaluated to assess their efficacy in killing mosquito vectors. These forms have been widely used in agriculture since 1990 to control pests. Research articles by Bond et al. (2004), Cetin et al. (2005), Sadanandane et al. (2009), Hertlein et al. (2010), Dos Santos Dias L (2017), Gimnig et al. (2020) and World Health Organization (WHO) reports from 2007, 2008, and 2011 [10–18] support the efficacy of these formulations in suppressing the larvae of *Aedes*, *Anopheles*, and *Culex* species in both laboratory and field settings.

Clarke Environmental Technologies (I) Ltd., a company based in Mumbai, India, has released three new formulations of Natular to accompany these changes: Extended Release Bi-Layer Tablet (7.48% DT), Emulsifiable Concentrate (20.6% EC), and Extended Release Granule (2.5% DT). The formulations of Natular contain Spinosad as their active ingredient and are developed for evaluation against mosquito vectors in diverse ecological settings.

In Puducherry, India, the Indian Council of Medical Research-Vector Control Research Centre (ICMR-VCRC) assessed the efficacy of these formulations against *Anopheles stephensi* Liston immature stages in Phase I (laboratory) and Phase II (small-scale field) evaluations. The objective of the present study was to evaluate the efficacy of these formulations against the urban malaria vector, *An. stephensi*, in Chennai, Tamil Nadu, India, via large-scale (Phase III) field trials. The trials commenced in July 2021 and concluded in August 2022 in accordance with the sampling protocols specified in the WHO Pesticide Evaluation Scheme [19].

## Materials and methods

### Test materials

Spinosad, a tetracyclic macrolide compound, is composed of two metabolites: $C_41H_{65}NO_{10}$ (spinosyn A) and $C_{42}H_{67}NO_{10}$ (spinosyn D). The actinomycete-class soil bacterium *S. spinosa* is responsible for the synthesis of this compound. Spinosad, acting as a nicotinic agonist, disrupts the functionality of ion channels regulated by GABA and nicotinic acetylcholine receptors (AChRs), resulting in the depolarisation and subsequent excitation of neurons in insects [20].

In India, three formulations of Natular (7.48% DT, 20.6% EC, and 2.5% G30) were supplied by the manufacturer, Clarke Environmental Technologies (I) Ltd., located in Mumbai, India, for the large-scale (Phase III) evaluation against *An. stephensi*.

Natular DT, a tablet formulation, featurestwo-layer design that includes an effervescent layer that immediately releases the active ingredient Spinosad into breeding habitats, quickly killing existing mosquito larvae. The second layer slowly releases into the water column over a period up to 60 days. Natular G30, granular formulation, is a multi-brood, extended-release granule formulation that releases adequate levels of Spinosad for up to 30 days. Natular EC, emulsifiable concentrate, a single-brood liquid formulation, releases adequate levels of Spinosad and is applied on a 7-day interval [21].

### Study area

The large-scale (Phase III) evaluation of Natular formulations was carried out in Chennai, Tamil Nadu, India, under the jurisdiction of the Greater Chennai Corporation (GCC). Chennai is geographically located between 12°50'49" N and 13°17'24" N, and longitude between 79°59'53" E and 80°20'12" E, along the Coromandel Coast in the northern region of Tamil Nadu [22]. Presently, the estimated population of the city is 7.1 million, distributed among three regions: North Chennai, Central Chennai, and South Chennai. The city encompasses a total area of 426 square kilometers [22]. Additionally, these areas are divided into fifteen zones, comprising a total of 200 wards. The formulations of Natular assessed in Phase III focused on Zones I (Tiruvottiyur), IV (Tondiarpet), and V (Royapuram) in North Chennai (Fig 1). Approval for conducting the trial was obtained from the Chief Vector Control Officer, Greater Chennai Corporation (GCC) and involved as one of the field-site investigators in the execution and completion of the evaluation.

High temperatures and high levels of humidity are characteristic features of Chennai's tropical wet and dry climate for most of the year. Due to its littoral and equatorial location, the city experiences minimal fluctuations in seasonal temperatures. Minimum temperatures in January range between 18 and 20 degrees Celsius, whereas maximum temperatures in May and June reach 38–42 degrees Celsius [22]. The northeast monsoon, which occurs between mid-October and mid-December, accounts for the majority of the annual precipitation, approximately 1400 millimeters [22].

The most frequently reported vector-borne diseases in Chennai on an annual basis are malaria and dengue. Malaria transmission is perennial, exhibiting its highest incidence rates from July to October. Annually, between 53.6% and 78.8% of the state's malaria cases have been reported in the city alone over the last two decades [23]. As of October 2017, 3,303 of the total 4,671 malaria cases in the state were reported in Chennai. Chennai documented a total of 2823 malaria cases in 2018 and 1452 malaria cases in 2019, with *Plasmodium vivax* constituting the prevailing parasite species, accounting for 95–97% of the reported cases [23].

Chennai has witnessed the implementation of numerous housing, slum development, and rehabilitation programs by the Tamil Nadu Urban Habitat Development Board. Zones I, IV, and V of North Chennai in particular contain a significant number

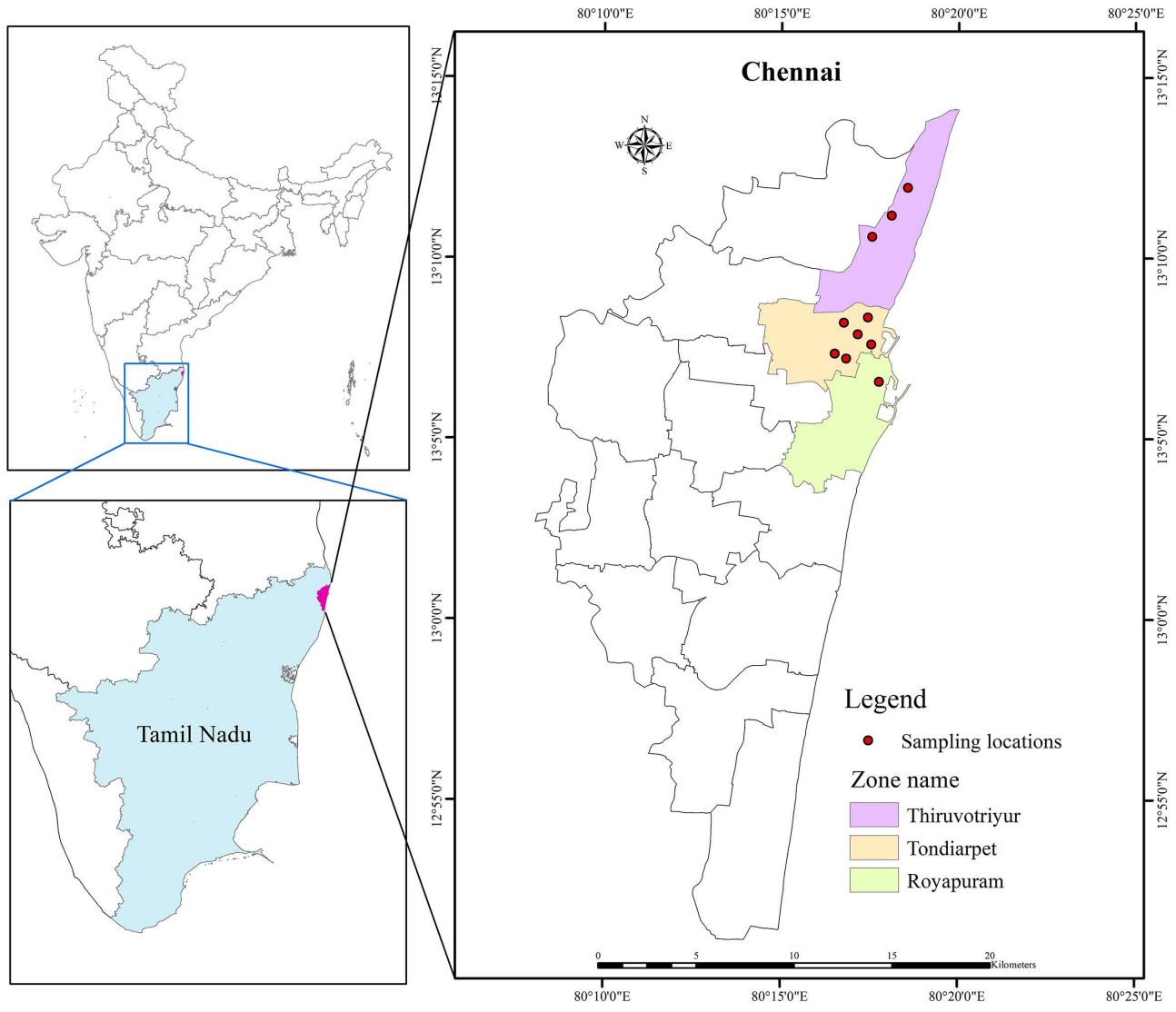

**Fig 1. Map of Chennai city (GCC) showing study zones and divisions.**

of Housing Board tenements and tsunami rehabilitation accommodations. The residents of these tenements and quarters must rely on Chennai Metrowater to deliver water *via* tanker trucks; however, due to the lack of piped water infrastructure. From the 200-liter plastic containers in which residents store this water, they pump it to the 500-liter Sintex overhead tanks (OHTs) located on the first through third floors. Unfortunately, the presence of these overhead containers facilitates the proliferation of *An. stephensi*, the vector of urban malaria. Despite the implementation of regular control strategies, such as LSM and weekly temephos applications in OHTs, malaria remains a substantial public health issue within the city's limits.

## Study design

In 500-liter overhead (sintex) tanks (OHTs), Natular formulations were evaluated against *An. stephensi* at the optimum field application dosage determined in the phase II trial [ICMR-VCRC, unpublished data]. The optimum application

dosages determined in the phase II trial for different formulations were one tablet/200L for 7.48% DT, 0.073 mg/L for 20.6% EC, and 0.074 mg/L for 2.5% G30. For evaluation, a total of 122 OHTs breeding with *An. stephensi* were selected. Seventy-six were randomly assigned to receive treatment with 20.6% EC [28 in winter (W), 24 in rainy weather (R), and 24 in summer (S)]. The remaining 46 (16 in R, 15 in W, and 15 in S) served as untreated controls. In trials containing 2.5% G30, 128 OHTs bred with *An. stephensi* were chosen. A total of 84 breeding habitats were assigned to receive the treatment (20 in R, 34 in W, and 30 in S); the remaining 44 breeding habitats (14 in R, 15 in W, and 15 in S) were designated as controls. For the 7.48% DT evaluation, 136 OHTs that were breeding with *An. stephensi* were chosen. A total of 83 individuals were subjected to treatment (20 in R, 15 in W, and 35 in S), while 53 were utilized as controls (23 in R, 15 in W, and 15 in S).

The optimal field application dosages of Natular formulations were applied to the selected habitats: 0.274 mg (AI)/L for 20.6% EC, 0.06 mg (AI)/L for 2.5% G30, and 1 tablet/200 L for 7.48% DT. After diluting the 20.6% EC formulation with water as needed, it was applied to the treatment group using a 2-liter hand compression sprayer from Foggers India Pvt. Ltd. that had a jet nozzle. A manual application was performed to equally distribute the 2.5% G30 granular formulation across all habitats. Similarly, the DT formulation was released manually into the larval habitats from the water surface.

In the designated habitats, larval and pupal densities were recorded twice weekly by drawing samples with enamel dippers with a capacity of 300 ml before treatment. To evaluate the effectiveness of the formulations, larval and pupal density in the treated habitats was monitored weekly from day one, day two, day three, and day seven after treatment, until the treated habitats reached a density level equivalent to that of the untreated habitats. After sampling and quantifying mosquito pupae and larvae by instar, the specimens were returned to their natural habitats. Monitoring immature densities required three samples from each habitat; on each sampling day, the pH and temperature of the habitat water were recorded. The data were recorded in accordance with the specified format, and observations were carried out for a minimum of two to three treatments per season for each formulation and species. Furthermore, the trials were replicated over the course of three different seasons.

## Data analysis

The data analysis involved calculating the average number of pupae and larvae collected per dip on each sampling day for each habitat, considering both the treatment and control groups. The third and fourth instar larvae were classified as late instar larvae (L3+L4), whereas the first and second instar larvae were grouped as early instar larvae (L1+L2).

To calculate the percentage reduction in the densities of early and late instar larvae and pupae on post-treatment days, Mulla's formula [24] was used as shown below:

$$\% \text{ Reduction } = 100 - [(C1 / T1) \times (T2/C2) \times 100]$$

Here, C1 and C2 represent the average number of larvae per dip in the control habitats before and after treatment. At the same time, T1 and T2 denote the number of larvae per dip in the treatment habitats before and after treatment, respectively. Using a two-way ANOVA, we assessed the arcsine-transformed percentage reduction between the formulations and post-treatment days for each season and re-treatment period.

## Results

### Efficacy of Natular 7.48% DT formulation against *An. stephensi*

**Rainy season.** Application of the 7.48% DT formulation at one tablet/200 L in overhead tanks with clean water resulted in a substantial reduction of *An. stephensi* pupae density by 98–100% for 8 weeks (Fig 2a). The treatment also effectively decreased the density of late and early instars by more than 90–100% up to week 8 post-treatment (Figs 2b and c). However, the efficacy gradually declined, prompting a retreatment at week 11 post-treatment. Remarkably, complete

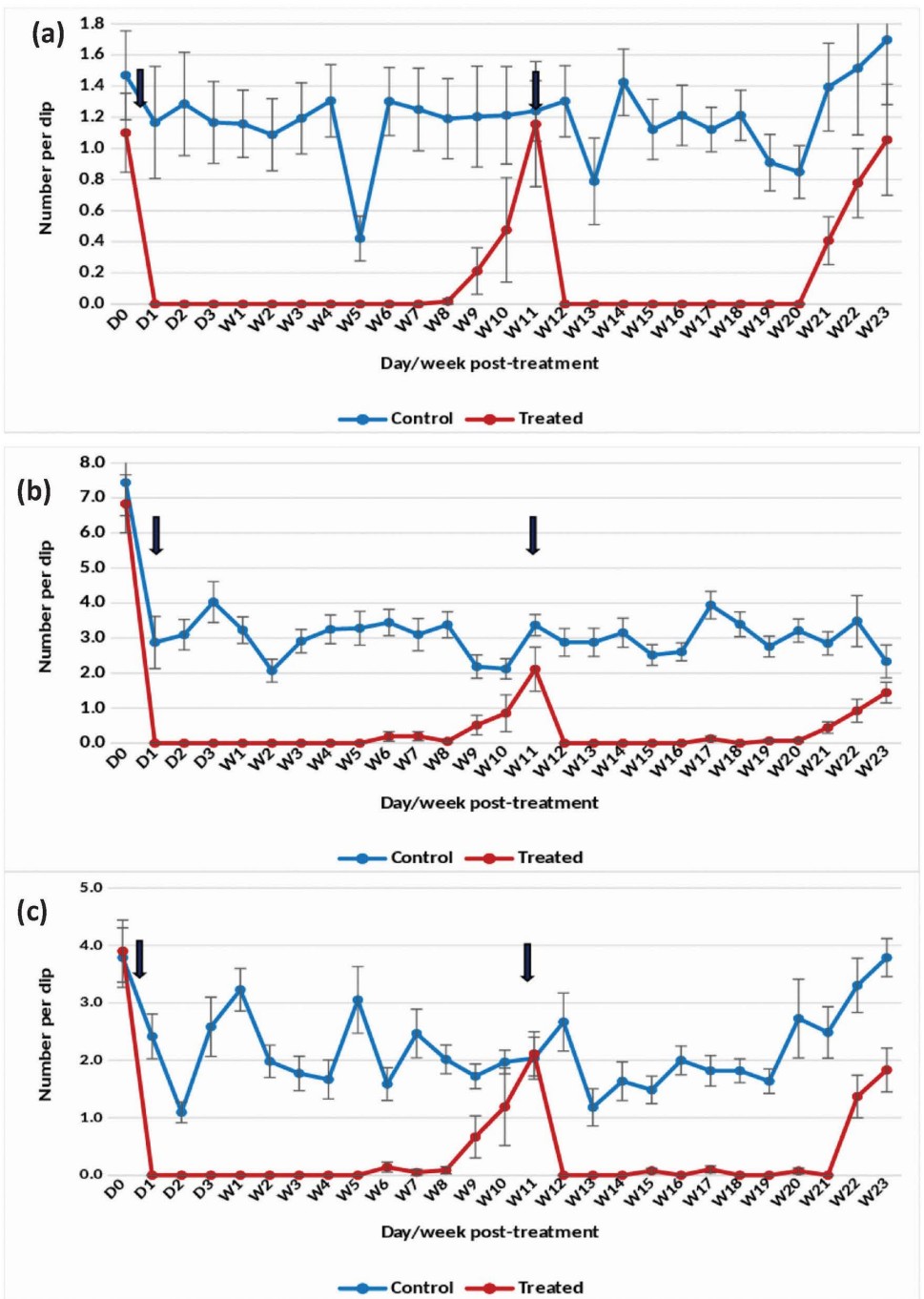

**Fig 2. Mean number of *An. stephensi* in untreated and treated overhead sintex tanks with Natular 7.48% DT at 1 tablet/200 L (Rainy)- (a) Pupae, (b) Late instar and (c) Early instar.**

(100%) suppression of pupal density was sustained for 9 weeks following retreatment. Retreatment at one tablet/200 L led to a persistent >80–100% reduction in late and early instar larval density for 10 weeks.

**Winter season.** In overhead tanks treated with the DT formulation at one tablet/200 L, the reduction in *An. stephensi* pupae and early instar density reached 100% for 9 weeks (Fig 3a and b). Complete (100%) suppression of late instar density was observed for 8 weeks, with a 93% reduction on week 9 post-treatment (Fig 3c). However, the density gradually increased, and by week 12 post-treatment, the reduction fell below 50%. Retreatment with the DT formulation resulted in a renewed 100% reduction in pupal density for 8 weeks. Late and early instar larval density was also reduced by >80–100% for 8 weeks following retreatment.

**Summer season.** Treatment with the DT formulation at one tablet/200 L in overhead tanks with *An. stephensi* breeding achieved an 85–100% reduction in pupal density for 8 weeks (Fig 4a). The treatment also effectively reduced the density of late and early instars by 86–100% up to week 8 post-treatment (Fig 4b and c). Following this period, pupal and larval densities increased, prompting a retreatment on week 11 post-treatment. After retreatment, a 100% reduction in pupal density was observed for 7 weeks. Retreatment also resulted in a sustained decrease of late instar larval density by 80–100% for 8 weeks and early instar density by 76–100% for 7 weeks.

### Efficacy of natular 2.5% G30 formulation against An. stephensi

**Rainy season.** The application of the Natular 2.5% G30 formulation at 0.074 mg (AI)/L in overhead tanks with clean water effectively suppressed the density of *An. stephensi* pupae by 91–100% up to week 4 post-treatment (Fig 5a). For late and early instars, the formulation resulted in a consistent 88–100% reduction over the same period (Fig 5b and c). However, in week 5 post-treatment, there was an increase in pupal and larval densities, prompting retreatment in week 6 post-treatment. Following retreatment, complete (100%) suppression of pupal density was sustained for 5 weeks. The retreatment also led to a reduction in late and early instar densities by more than 80–100% for 5 weeks after retreatment.

**Winter season.** In overhead tanks treated with the 2.5% G30 formulation, there was an 88–100% reduction in the density of *An. stephensi* pupae for 4 weeks at 0.074 mg (AI)/L (Fig 6a). The formulation was found to consistently reduce late and early instar larval density by >80–100% for 3 weeks (Fig 6b and c).

**Summer season.** At a concentration of 0.074 mg (AI)/L in overhead tanks with clean water, the G30 formulation achieved complete suppression of the density of *An. stephensi* pupae, maintaining a 100% reduction up to 3 weeks post-treatment (Fig 7a). The density of late and early instars was reduced by 84–100% for 3 weeks after treatment (Fig 7b and c). Subsequently, pupal and larval densities began to increase, leading to a retreatment on week 7 post-treatment. Following retreatment, the density of pupae and late and early instars was reduced by 85–100% for 4 weeks.

### Efficacy of natular 20.6% EC formulation against An. stephensi

**Rainy season.** The application of the 20.6% EC formulation at 0.073 mg (AI)/L in overhead tanks with *An. stephensi* breeding resulted in a 100% reduction in pupal density for 2 weeks (Fig 8a). By week 3 post-treatment, the reduction had slightly decreased to 77.64%. The reduction in the density of late and early instars remained consistently greater than 80–100% up to week 2 post-treatment (Fig 8b and c).

**Winter season.** In overhead tanks treated at 0.073 mg (AI)/L, the EC formulation achieved a 100% reduction in the density of *An. stephensi* pupae for 2 weeks (Fig 9a). The decrease in the density of late instars ranged from 96–100%, and for early instars, it was 85–100% for 2 weeks (Fig 9b and c). However, in weeks 3 and 4 post-treatment, larval and pupal densities gradually increased, reaching a level comparable to the control. Following retreatment in week 4 post-treatment, the reduction in the density of pupae and late instars was 100%, and for early instars, it was 98% for 1 week after retreatment.

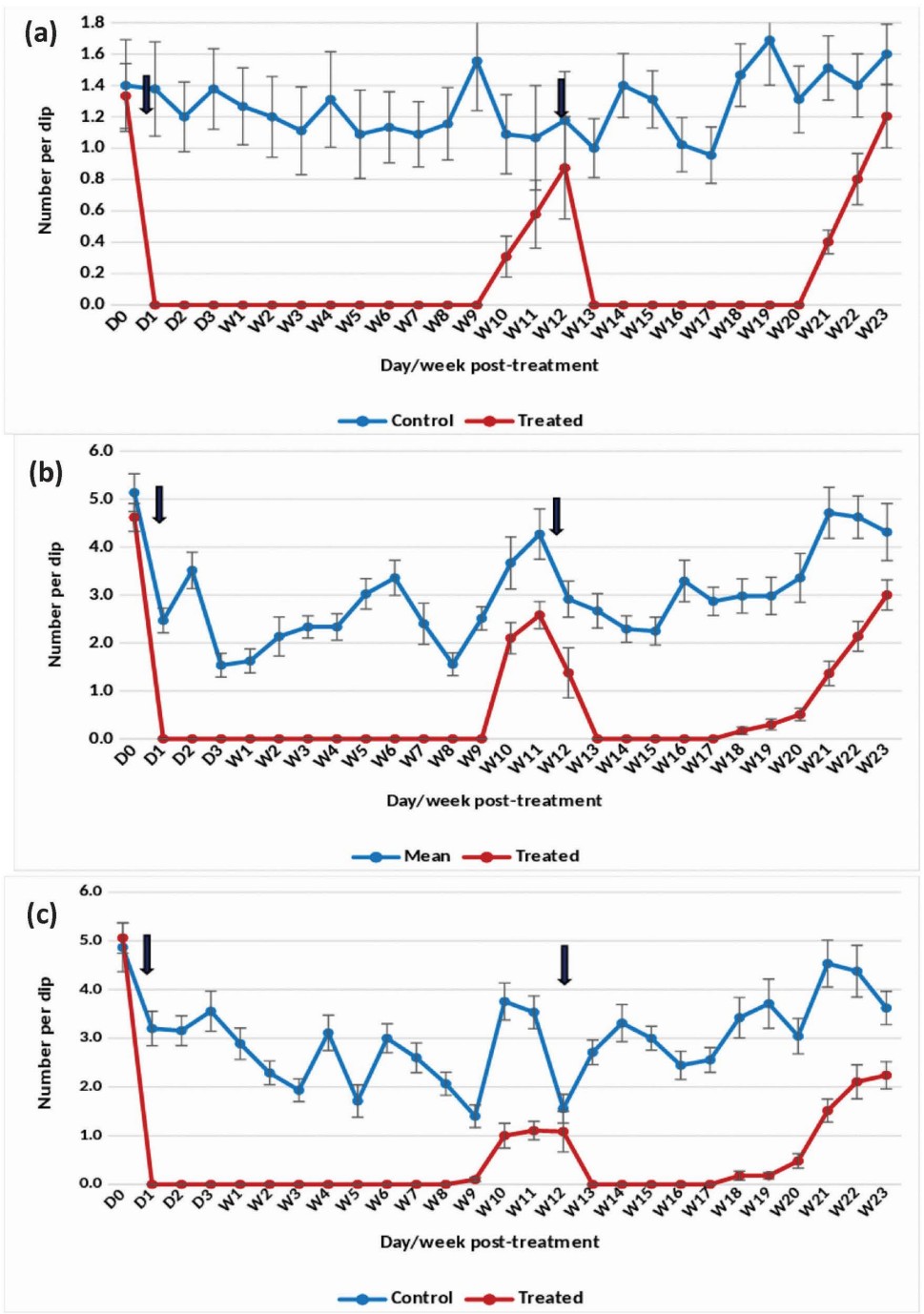

**Fig 3. Mean number of *An. stephensi* in untreated and treated overhead sintex tanks with Natular 7.48% DT at 1 tablet/200 L (Winter)- (a) Pupae, (b) Late instar and (c) Early instar.**

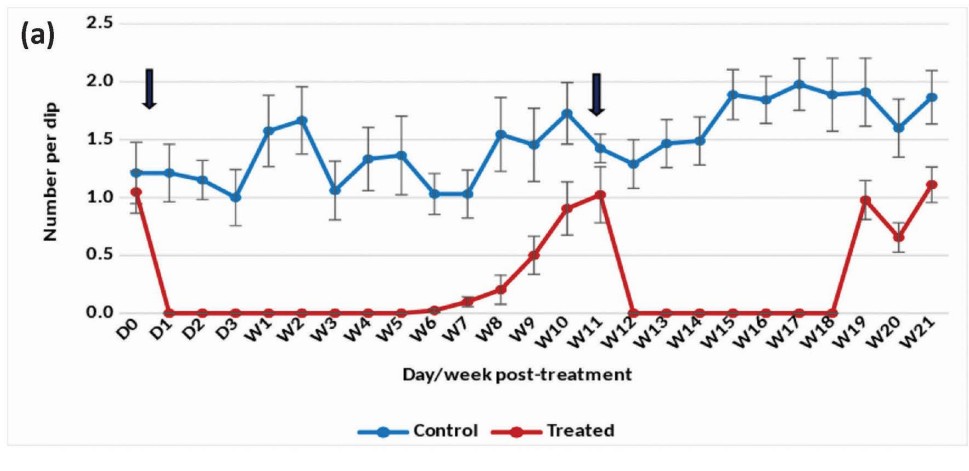

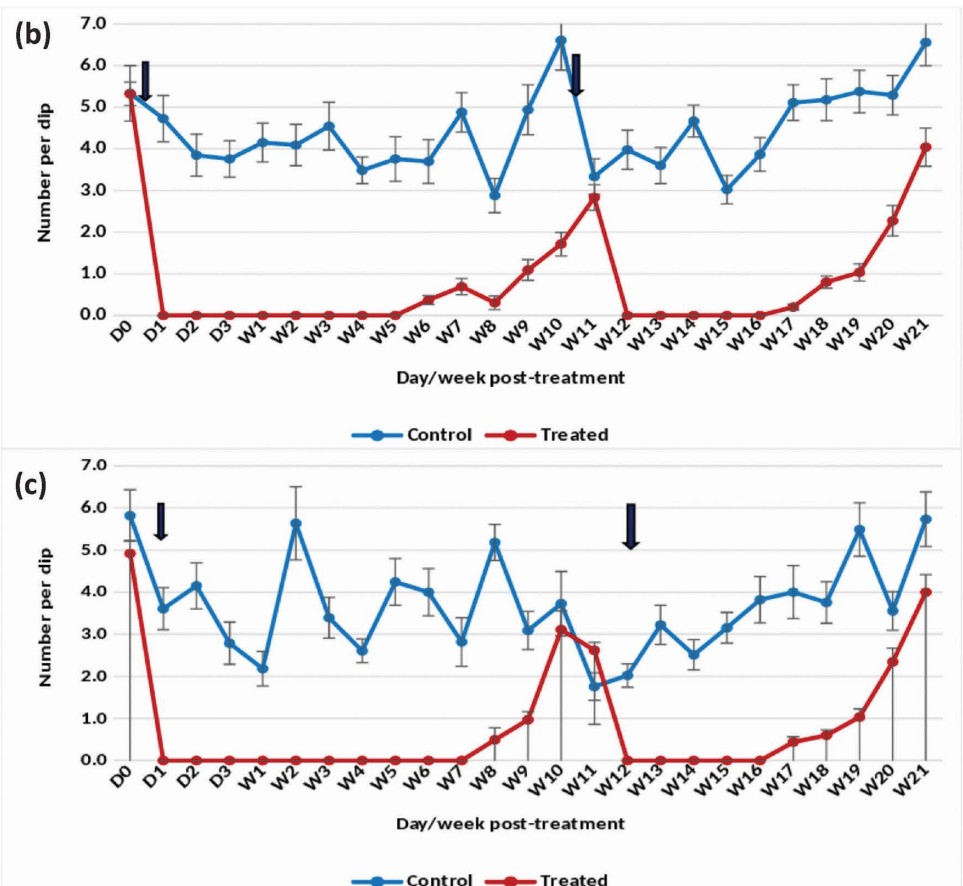

**Fig 4.** Mean number of *An. stephensi* in untreated and treated overhead sintex tanks with Natular 7.48% DT at 1 tablet/200 L (Summer)- (a) Pupae, (b) Late instar and (c) Early instar.

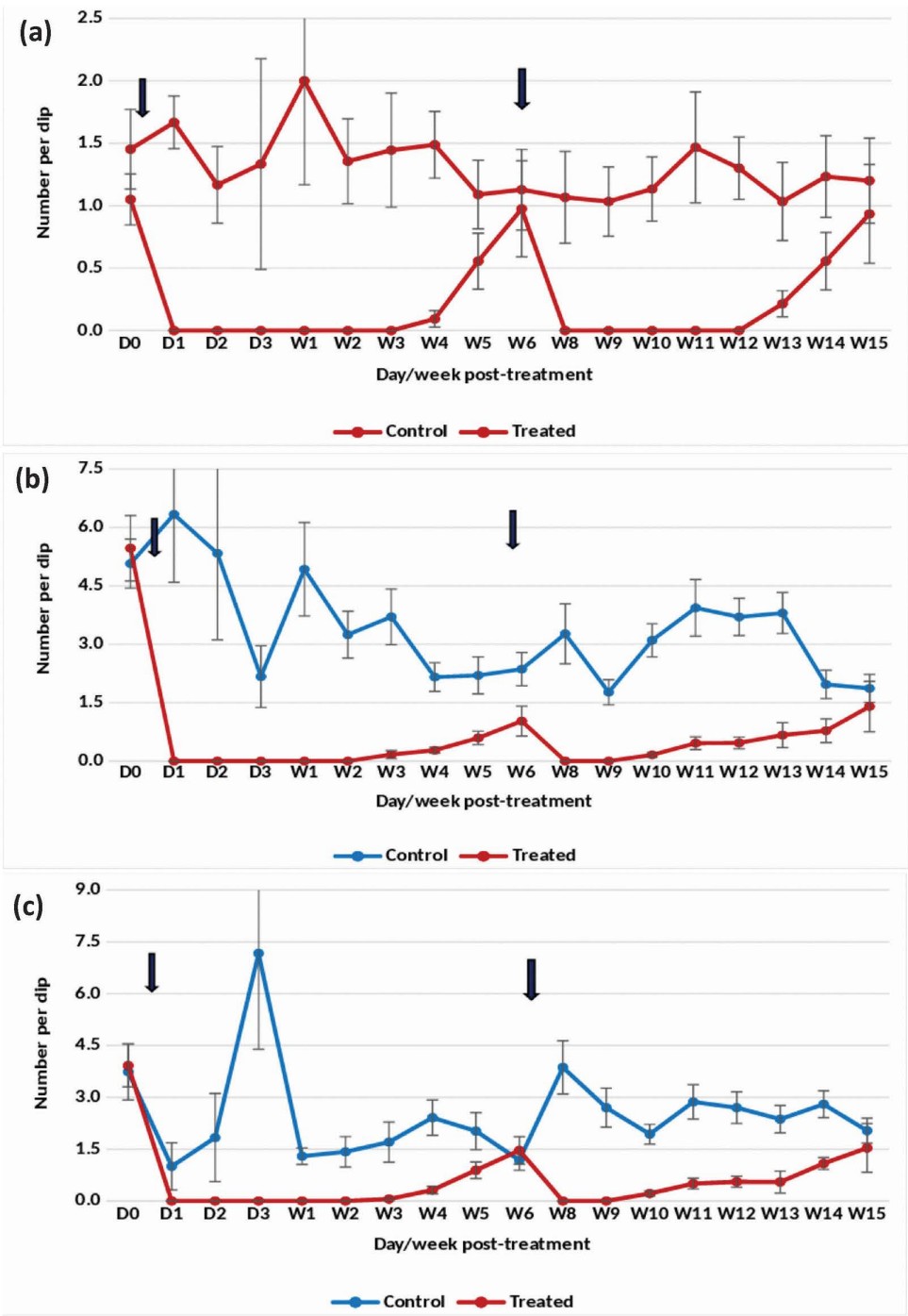

**Fig 5. Mean number of *An. stephensi* pupae in untreated and treated overhead sintex tanks with Natular 2.5% G30 at 0.074 mg (AI)/L (Rainy)- (a) Pupae, (b) Late instar and (c) Early instar.**

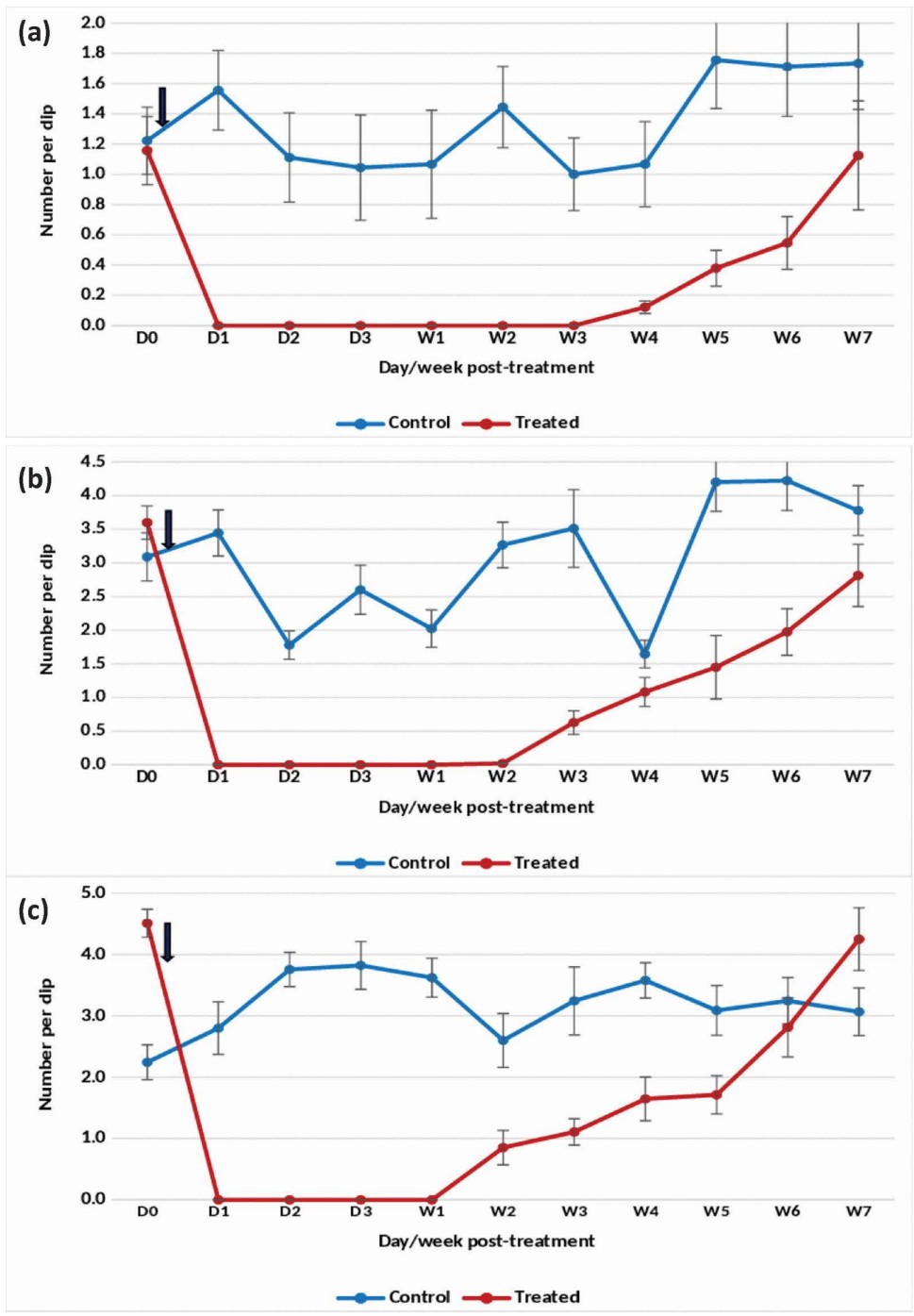

**Fig 6. Mean number of *An. stephensi* in untreated and treated overhead sintex tanks with Natular 2.5% G30 at 0.074 mg (AI)/L (Winter) – (a) Pupae, (b) Late instar and (c) Early instar.**

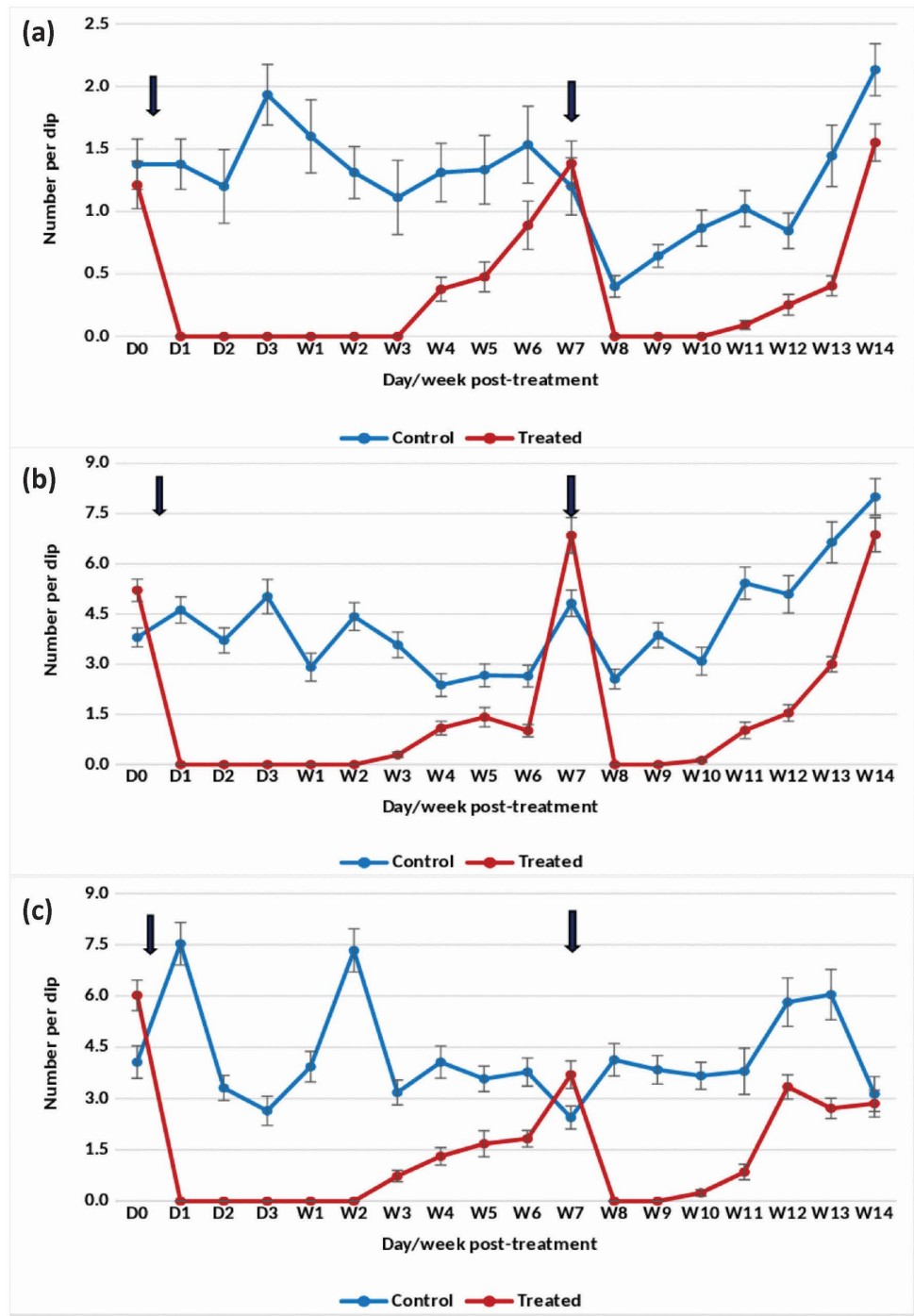

**Fig 7.** Mean number of *An. stephensi* in untreated and treated overhead sintex tanks with Natular 2.5% G30 at 0.074 mg (AI)/L (Summer) – (a) Pupae, (b) Late instar and (c) Early instar.

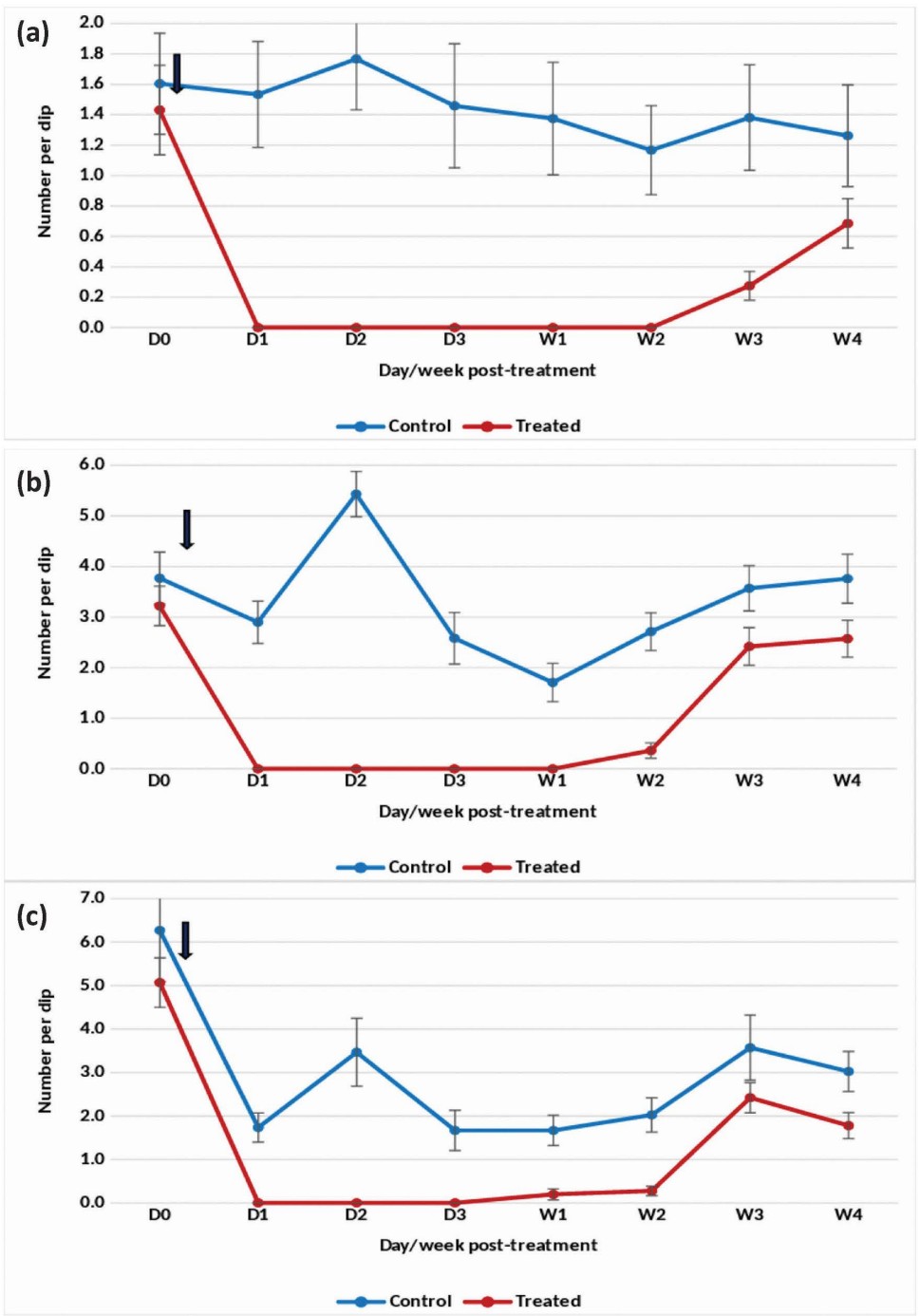

**Fig 8. Mean number of *Anopheles stephensi* in untreated and treated overhead sintex tanks with Natular 20.6% EC at 0.073 mg (AI)/L (Rainy)-(a) Pupae, (b) Late instar and (c) Early instar.**

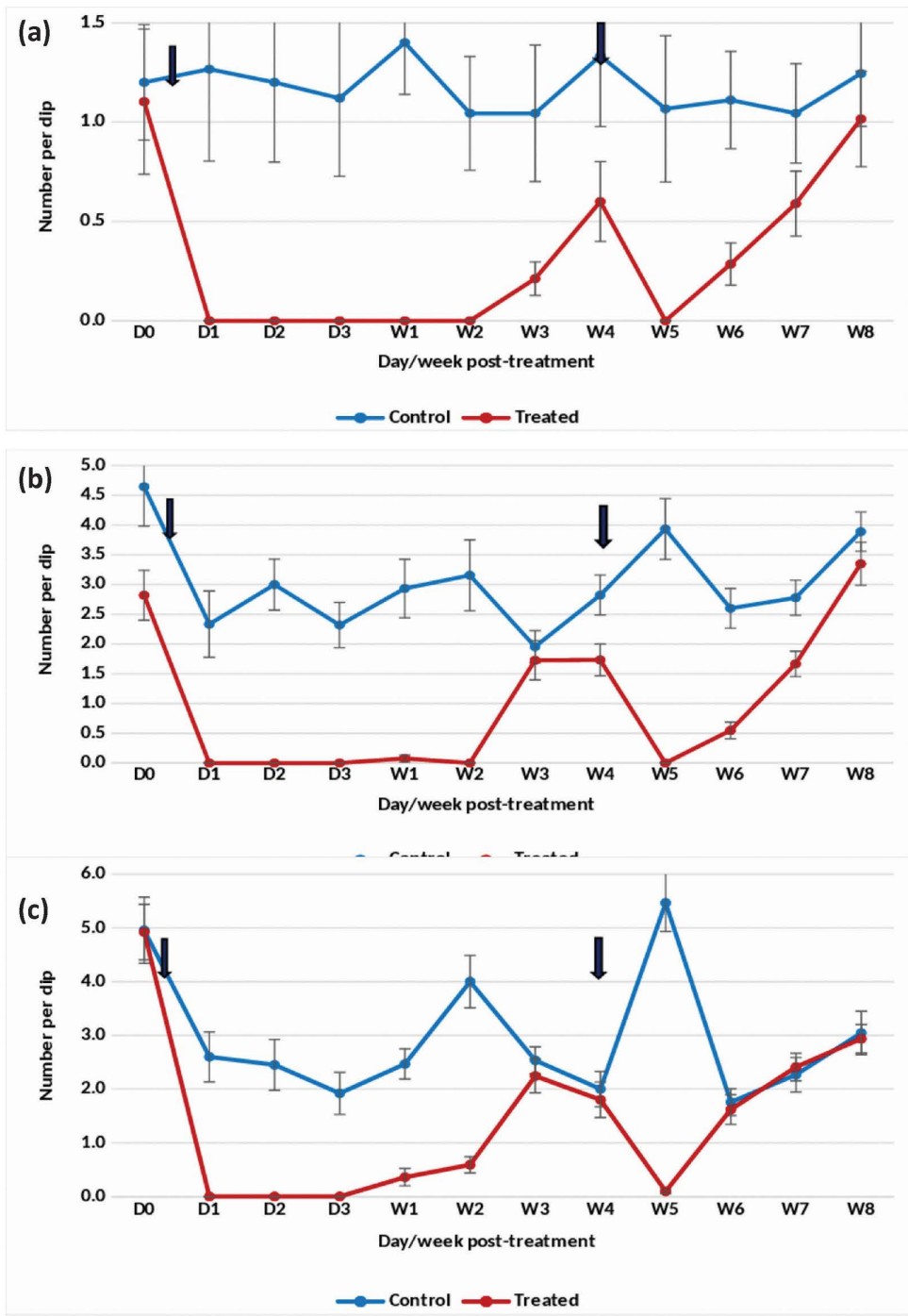

**Fig 9. Mean number of *Anopheles stephensi* in untreated and treated overhead sintex tanks with Natular 20.6% EC at 0.073 mg (AI)/L (Winter)- (a) Pupae, (b) Late instar and (c) Early instar.**

**Summer season.** The application of the 20.6% EC formulation at 0.073 mg (AI)/L in overhead tanks resulted in a 100% reduction in pupal and late instar larval density for 2 weeks after the first treatment (Fig 10a and b). In the case of early instars, a 90–100% reduction was observed for 2 weeks (Fig 10c). Subsequently, habitat retreatment was done on

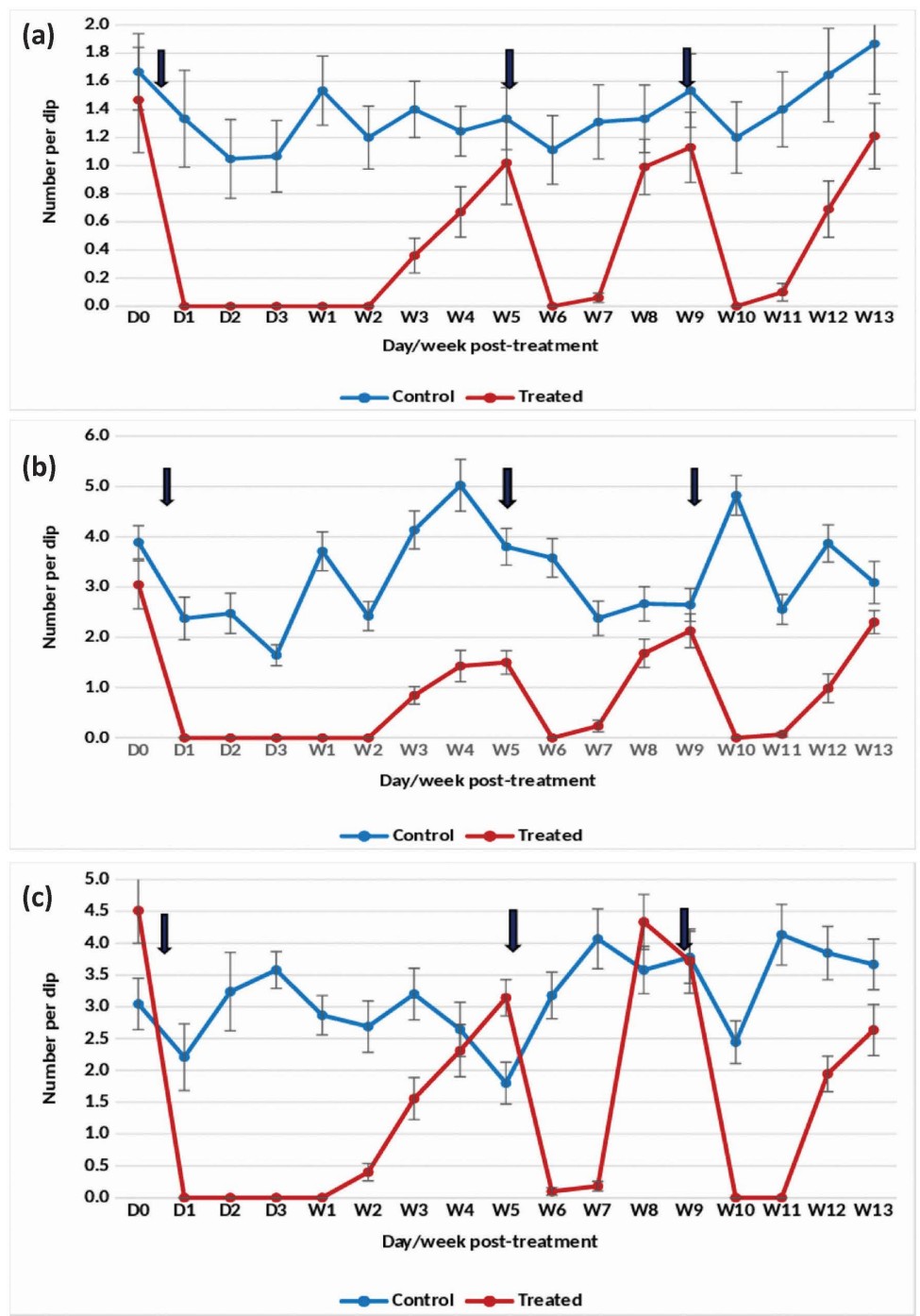

**Fig 10. Mean number of *Anopheles stephensi* in untreated and treated overhead sintex tanks with Natular 20.6% EC at 0.073 mg (AI)/L (Summer)-** (a) Pupae, (b) Late instar and (c) Early instar.

weeks 5 and 9 post-treatment. The density of pupae, late, and early instar larvae was reduced by more than 90% for 2 weeks after the second and third treatments.

## Statistical analysis

### Early instars.

*Rainy season:* The 20.6% EC, 2.5% G30, and 7.48% DT formulations maintained an 80% reduction in mean larval density for 2, 4, and 8 weeks, respectively. We observed a significant difference between the formulations ($F_{(2, 13)}$ = 5.36, P = 0.020), but found no significant difference between the days ($F_{(12, 13)}$ = 2.11, P = 0.098). Following re-treatments, the 2.5% G30 and 7.48% DT formulations sustained an 80% reduction for 6 and 8 weeks, respectively. We detected a significant difference between the two formulations ($F_{(1, 7)}$ = 8.89, P = 0.020), but found no difference between the days ($F_{(11, 7)}$ = 2.12, P = 0.164).

*Winter season:* The 7.48% DT formulation showed significantly longer residual efficacy than the other two formulations ($F_{(2, 14)}$ = 6.00, P = 0.013). We also observed a significant difference between the days ($F_{(13, 14)}$ = 2.66, P = 0.040).

*Summer season:* The 20.6% EC, 2.5% G30, and 7.48% DT formulations maintained an 80% reduction in mean larval density for 2, 3, and 8 weeks, respectively. We found significant differences between the formulations ($F_{(2, 14)}$ = 5.00, P = 0.023) and the days ($F_{(11, 14)}$ = 3.08, P = 0.025). After the re-treatments, the 20.6% EC, 2.5% G30, and 7.48% DT formulations maintained an 80% reduction for 2, 4, and 8 weeks, respectively. We observed a significant difference between the formulations ($F_{(2, 8)}$ = 5.77, P = 0.028), but found no significant difference between the days ($F_{(9, 8)}$ = 3.15, P = 0.060).

### Late instars.

*Rainy season:* The 20.6% EC, 2.5% G30, and 7.48% DT formulations maintained an 80% reduction in mean larval density for 2, 4, and 8 weeks, respectively. The formulations showed a significant difference ($F_{(2, 13)}$ = 3.93, P = 0.046), but the days demonstrated no significant difference ($F_{(12, 13)}$ = 1.80, P = 0.152). Following re-treatments, the 2.5% G30 and 7.48% DT formulations maintained an 80% reduction for 3 and 9 weeks, respectively. We observed a significant difference between the two formulations ($F_{(1, 6)}$ = 7.93, P = 0.030), but found no difference between the days ($F_{(11, 6)}$ = 2.70, P = 0.117).

*Winter season:* The 7.48% DT formulation had a much longer residual effect than the other two formulations ($F_{(2, 13)}$ = 5.90, P = 0.015), but there was no significant difference between the days ($F_{(13, 13)}$ = 1.79, P = 0.154). The 2.5% G30 and 7.48% DT formulations maintained an 80% reduction for 1 and 7 weeks after re-treatments, respectively. We observed no significant difference between these formulations ($F_{(1, 2)}$ = 2.75, P = 0.239) or between the days ($F_{(10, 2)}$ = 0.99, P = 0.602).

*Summer season:* The 20.6% EC, 2.5% G30, and 7.48% DT formulations maintained an 80% reduction in mean larval density for 2, 3, and 8 weeks, respectively. We found significant differences between the formulations ($F_{(2, 14)}$ = 3.65, P = 0.050) and between the days ($F_{(12, 14)}$ = 3.60, P = 0.013). The 20.6% EC, 2.5% G30, and 7.48% DT formulations maintained an 80% reduction for 1, 4, and 6 weeks following re-treatments. We observed significant differences between the formulations ($F_{(2, 7)}$ = 5.21, P = 0.041), but found no difference between the days ($F_{(8, 7)}$ = 8.03, P = 0.006).

### Pupae.

*Rainy season:* The 20.6% EC, 2.5% G30, and 7.48% DT formulations maintained an 80% reduction in mean larval density for 2, 4, and 8 weeks, respectively. We found no significant difference between the formulations ($F_{(2, 13)}$ = 1.98, P = 0.178) or the days ($F_{(12, 13)}$ = 2.08, P = 0.102). Following re-treatments, the 2.5% G30 and 7.48% DT formulations maintained an 80% reduction for 5 and 9 weeks, respectively. We found no significant difference between these formulations ($F_{(1, 7)}$ = 3.49, P = 0.104) or between the days ($F_{(11, 7)}$ = 1.77, P = 0.229).

*Winter season:* Despite no significant difference between the formulations (F (2, 14) = 3.31, P = 0.067) or the days (F (13, 14) = 2.41, P = 0.057), the 7.48% DT formulation demonstrated higher residual efficacy compared to the other two formulations. Following re-treatments, the 2.5% G30 and 7.48% DT formulations maintained an 80% reduction for 1 and 8 weeks, respectively. We found no significant differences between the formulations (F (1,1) = 1.00, P = 0.500) or between the days (F (9,1) = 0.86, P = 0.692).

*Summer season:* The 20.6% EC, 2.5% G30, and 7.48% DT formulations maintained an 80% reduction in mean larval density for 2, 3, and 8 weeks, respectively. We observed no significant difference between the formulations (F (2, 14) = 3.19, P = 0.072), but we found a significant difference between the days (F (12, 14) = 3.23, P = 0.020). Following re-treatments, the 20.6% EC, 2.5% G30, and 7.48% DT formulations sustained an 80% reduction for 2, 4, and 7 weeks. We found no significant differences between the formulations (F (2, 8) = 3.56, P = 0.078), nor between the days (F (9, 8) = 2.79, P = 0.081).

## Discussion

*Anopheles stephensi* is primarily responsible for transmitting malaria in urban areas of India and beyond [25–28]. Even at low population densities, this mosquito species is capable of transmitting both *P. vivax* and *P. falciparum* [29]. *An. stephensi* breeds primarily in sumps, cement tanks, wells, and overhead tanks that contain water. To mitigate the transmission of malaria caused by *An. stephensi* in urban regions, the urban malaria scheme of the National Centre for Vector Borne Diseases Control (NCVBDC) in India utilizes chemical and biological insecticides, as well as larvivorous fish, for controlling larval populations. Currently, weekly applications of Temephos 50% EC (an organophosphate insecticide) and Bti (WP and 12 AS) formulations are utilized as larvicides in order to control the larval population. Nevertheless, previous studies have documented temephos resistance [30]. Besides, chemical insecticides present health and environmental risks. To avoid these situations, the biolarvicide Spinosad (Natular) can be used as an appropriate replacement for traditional chemical larvicides. The World Health Organization Pesticide Evaluation Scheme (WHOPES) has evaluated several forms of Spinosad to assess their effectiveness in killing mosquito larvae in various countries [16–18]. These forms include 8.33% DT, 0.5% GR, and 12% SC. In a recent publication, Clarke Environmental Technologies (I) Ltd., based in Mumbai, India, introduced three novel Spinosad (Natular) formulations: 7.48% DT, 20.6% EC, and 2.5% G30 [12,15–18]. These formulations demonstrated efficacy against *An. stephensi* larvae in both simulated field (phase II) and laboratory (phase I) conditions (ICMR-VCRC, unpublished report).

The statistical analysis indicated significant variation in the residual efficacy of the three Natular® formulations in different seasons. 7.48% DT formulation consistently demonstrated a higher and longer reduction in larval density of early instars (Figs 2c, 3c & 4c), compared to the 2.5% G30 and 20.6% EC formulations (Figs 5c, 6c, 7c, 8c, 9c, & 10c). The significant differences observed between formulations (P < 0.05) during the rainy, winter, and summer seasons confirm that the formulation type had a measurable effect on larval reduction. However, the differences with respect to sampling days were not statistically significant (P > 0.05), indicating that once applied, and each formulation maintained a relatively stable level of efficacy during its active period. The 7.48% DT formulation, in particular, showed extended residual activity of up to 8 weeks in all seasons (Figs 2c, 3c & 4c), as reflected by consistently high larval reduction and significant formulation effects in the ANOVA results. The lack of significant seasonal interaction effects in some comparisons further supports the strength of this formulation in different seasons. Overall, the statistical analysis indicate, that the characteristics of the formulation, rather than temporal variation, influenced the larvicidal effect, with 7.48% DT formulation as effective and operationally sustainable option for *An. stephensi* larval control.

The analysis of late instar data indicated efficacy of Spinosad varied significantly between the three Natular® formulations in different seasons. Overall, the 7.48% DT formulation consistently showed prolonged residual activity, maintaining more than 80% reduction in larval density for up to 8–9 weeks (Figs 2b, 3b & 4b), compared with 2–4 weeks for the 2.5% G30 and 20.6% EC formulations (Figs 5b, 6b, 7b, 8b, 9b, & 10b). The significant differences observed between

formulations (P < 0.05) indicate that formulation type have an influence on larval mortality and residual efficacy. On the other hand, comparisons between days within each trial showed no significant difference (P > 0.05), indicating that once applied, and each formulation maintained a consistent level of larvicidal activity during its effective period. However, during the summer, significant differences were observed both between formulations and days, perhaps reflecting the influence of higher temperatures and the faster degradation of the active ingredient in the larval habitats.

Following re-treatments, the 7.48% DT formulation again showed extended persistence, sustaining a reduction in larval densities for up to 9 weeks during the rainy and 7 weeks during the winter seasons, while the efficacy of the other formulations decreased more quickly (Figs 2-4). These results together indicate that the DT formulation offers a more long-lasting and consistent performance under fluctuating seasonal conditions. Based on the statistical analysis, the significant F-values and corresponding P-values (<0.05) indicate a statistically significant difference in efficacy among formulations, confirming that formulation characteristics, particularly the mode of release of Spinosad, play a crucial role in determining the duration and stability of larvicidal action.

The analysis of pupal density reduction showed that, although the differences among the three Natular® formulations were not statistically significant (P > 0.05) in most trials, the 7.48% DT formulation (Figs 2a, 3a & 4a) consistently showed superior and extended residual efficacy compared to the 2.5% G30 (Figs 5a, 6a & 7a) and 20.6% EC formulations (Figs 8a, 9a & 10a). The lack of statistically significant variation between formulations suggests quantitative differences were observed, and the variability within replicates was comparatively high or the overall effect sizes were modest. Nevertheless, the observed trend in different seasons constantly indicated the DT formulation, which maintained an 80% reduction for up to eight to nine weeks, indicating its superior efficacy and more consistent larvicidal activity on the pupal development (Figs 2a, 3a & 4a).

The absence of significant differences over sampling days (P > 0.05) indicates that once the formulations were applied, their pupicidal effects remained comparatively stable throughout their effective duration. The significant effect observed during the summer season (P = 0.020) possibly reflects the influence of environmental factors, such as raise in temperatures and organic content, could alter the degradation level of the active ingredient and affect the metamorphosis from larval-to-pupal stages.

Although the differences among formulations did not reach statistical significance, the uniform bio-efficacy of the 7.48% DT formulation through seasons highlights its operational advantage for field application. Its extended residual activity, under variable environmental conditions, suggests that this formulation could reduce the frequency of re-application required in larval control programs. Overall, the statistical analysis indicated that while all formulations were effective in lowering pupal densities, the DT formulation provided consistent and sustained efficacy.

During the large-scale (phase III) assessment conducted in Chennai city, the 20.6% formulation showed efficacy against *An. stephensi* for a duration of two weeks. At a dose of 0.073 mg (AI)/L administered every two weeks, it can be used as a control. At a concentration of 0.074 mg (AI)/L for four weeks, the 2.5% G30 formulation was efficacious against this species for four to five weeks in all seasons, making it suitable for controlling *An. stephensi*. With one exception, the 7.48% DT formulation demonstrated efficacy against *An. stephensi* for a duration of 8–9 weeks, irrespective of the season. Hence, the 7.48% DT formulation may be utilized to control *An. stephensi* larvae at a rate of one tablet per 200L for eight-week intervals.

## Conclusion

The WHO has designated Spinosad 7.48% DT as a viable vector control agent in potable water, specifically against container-breeding mosquitoes. When using formulations of Natular 20.6% EC, 2.5% G30, and 7.48% DT, the residual activity was found to be 2, 4, and 8 times greater, respectively, when compared to temephos. As a result of its novel mechanism of action as a nicotinic agonist, Spinosad has demonstrated no cross-resistance to chemical larvicides currently in use. Furthermore, in line with the Integrated Vector Management program, natular (Spinosad) formulations appear to be

effective biolarvicides for managing *An. stephensi*, as evidenced by their longer residual activity, efficacy, and favorable toxicological and environmental profile.

## Acknowledgments

We gratefully acknowledge the Director General of the Indian Council of Medical Research for approving the study. The authors acknowledge the technical assistance in the field rendered by Mr. R. Hemasankar, Senior Technician-2 and all project personnel employed in the study. We also acknowledge the staff of Greater Chennai Corporation for their assistance during the field study.

## Author contributions

**Conceptualization:** Candasamy Sadanandane, Ananganallur Nagarajan Shriram, Ashwani Kumar.

**Data curation:** Annamalai Sakthivel.

**Formal analysis:** Candasamy Sadanandane, Ananganallur Nagarajan Shriram.

**Funding acquisition:** Ananganallur Nagarajan Shriram, Manju Rahi.

**Investigation:** Mariapillai Kalimuthu, Ranganathan Krishnamoorthi, S Selvakumar, Annamalai Sakthivel.

**Methodology:** Candasamy Sadanandane.

**Project administration:** Ananganallur Nagarajan Shriram, Manju Rahi.

**Resources:** Ashwani Kumar, S Selvakumar.

**Supervision:** Ananganallur Nagarajan Shriram.

**Writing – original draft:** Candasamy Sadanandane, Mariapillai Kalimuthu, Ranganathan Krishnamoorthi, S Selvakumar, Annamalai Sakthivel.

**Writing – review & editing:** Ananganallur Nagarajan Shriram, Ashwani Kumar, Manju Rahi.

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
