## [Decision Letter · Decision Letter 0]

5 Mar 2024

Dear Dr. SHRIRAM,

Thank you for submitting your manuscript to PLOS ONE. After careful consideration, we feel that it has merit but does not fully meet PLOS ONE’s publication criteria as it currently stands. Therefore, we invite you to submit a revised version of the manuscript that addresses the points raised during the review process.

Please submit your revised manuscript by Apr 19 2024 11:59PM. If you need more time than this to complete your revisions, please reply to this message or contact the journal office at plosone@plos.org . A rebuttal letter that responds to each point raised by the academic editor and reviewer(s). You should upload this letter as a separate file labeled 'Response to Reviewers'.A marked-up copy of your manuscript that highlights changes made to the original version. You should upload this as a separate file labeled 'Revised Manuscript with Track Changes'.An unmarked version of your revised paper without tracked changes. You should upload this as a separate file labeled 'Manuscript'.

We look forward to receiving your revised manuscript.

Kind regards,

Yash Gupta, Ph.D.

Academic Editor

PLOS ONE

Journal Requirements:

3. In the online submission form, you indicated that "The data is available in the manuscript. However, the data would be made available upon request."

4. We note that Figure 1 in your submission contain map/satellite images which may be copyrighted. All PLOS content is published under the Creative Commons Attribution License (CC BY 4.0), which means that the manuscript, images, and Supporting Information files will be freely available online, and any third party is permitted to access, download, copy, distribute, and use these materials in any way, even commercially, with proper attribution. For these reasons, we cannot publish previously copyrighted maps or satellite images created using proprietary data, such as Google software (Google Maps, Street View, and Earth). For more information, see our copyright guidelines: http://journals.plos.org/plosone/s/licenses-and-copyright.

Additional Editor Comments:

Authors need to address orphan claims in the discussion sections with fig number or cited reference. The stats applied needs to be explained in terms of significance. Optionally, a more mechanistic insight into the observed phenomenon seems to be lacking in the discussion section. The manuscript needs revision according to the expert reviewers. There are no suggested further/revised experiments, and therefore a minor revision is recommended.

Reviewers' comments:

Reviewer's Responses to Questions

**Comments to the Author**

1. Is the manuscript technically sound, and do the data support the conclusions?

Reviewer #1: Yes

Reviewer #2: Yes

2. Has the statistical analysis been performed appropriately and rigorously?

Reviewer #1: I Don't Know

Reviewer #2: Yes

3. Have the authors made all data underlying the findings in their manuscript fully available?

Reviewer #1: Yes

Reviewer #2: Yes

4. Is the manuscript presented in an intelligible fashion and written in standard English?

Reviewer #1: Yes

Reviewer #2: Yes

Reviewer #1: The manuscript needs to be revised to improve it. The title needs to be more concise, the abstract needs to correctly capture the findings and statements of statistical significance of means of populations will be important. There are also certain questions that have to be addressed probably in the discussion section. A more detailed file with comments to the authors is attached.

Reviewer #2: The study titled “A large-scale field trial in Chennai, India found that Natular®, a biolarvicide, was effective in reducing populations of Anopheles stephensi, a major malaria vector” investigates the evaluation of effectiveness of three newly formulated biolarvicides, Natular 20.6% Emulsifiable Concentrate (EC), 2.5% G30 granular, and 7.48% tablet for direct application (DT), against Anopheles stephensi. The research aims to assess the best formulation and the dosage to make recommendations for better larval management of Anopheles spps. The finding of the study and the recommendation makes the study of high impact to control the diseases like malaria.

The insecticide resistance against the available antilarval insecticide Temephos calls for urgent action in vector control strategies to combat the vector in India. Insecticide resistance in the diseases vectors poses a significant threat to public health. This study provides an alternative solution using the new formulations with up to 100% larval control. These Natular formulations serve as potent bio larvicides, offering a valuable tool for An. stephensi control within the Integrated Vector Management (IVM) program. Development of new formulations is crucial for developing effective vector control strategies and preventing the spread of malaria parasites.

I think that this is a very nice and important contribution to the existing literature offering an alternative to insecticide resistance in An. stephensi. It should be published once the following issues have been addressed.

The authors don’t discuss the mechanism of action for the three different formulations.

Have any cross-resistance studies been conducted for the formulations.

**Do you want your identity to be public for this peer review?** For information about this choice, including consent withdrawal, please see our Privacy Policy

Reviewer #1: No

Reviewer #2: **Yes: ** RBS Kushwah

---

## [Author Response · Author response to Decision Letter 1]

20 Dec 2024

Journal Requirements:

Sl. No. Comments Clarifications/Answers

1 Please ensure that your manuscript meets PLOS ONE's style requirements, including those for file naming. The PLOS ONE style templates can be found at

Thank you for your guidance. We have carefully reviewed the PLOS ONE style requirements and have made the necessary modifications to the revised manuscript.

2 In your Methods section, please provide additional information regarding the permits you obtained for the work. Please ensure you have included the full name of the authority that approved the field site access and, if no permits were required, a brief statement explaining why. Thank you for your suggestion. We have incorporated the necessary information regarding permissions into the Methodology section in the revised manuscript/

3 In the online submission form, you indicated that "The data is available in the manuscript. However, the data would be made available upon request."

This policy applies to all data except where public deposition would breach compliance with the protocol approved by your research ethics board. If your data cannot be made publicly available for ethical or legal reasons (e.g., public availability would compromise patient privacy), please explain your reasons on resubmission and your exemption request will be escalated for approval. All the data is available in the manuscript itself.

4 We note that Figure 1 in your submission contain map/satellite images which may be copyrighted. All PLOS content is published under the Creative Commons Attribution License (CC BY 4.0), which means that the manuscript, images, and Supporting Information files will be freely available online, and any third party is permitted to access, download, copy, distribute, and use these materials in any way, even commercially, with proper attribution. For these reasons, we cannot publish previously copyrighted maps or satellite images created using proprietary data, such as Google software (Google Maps, Street View, and Earth). For more information, see our copyright guidelines: http://journals.plos.org/plosone/s/licenses-and-copyright. We require you to either (a) present written permission from the copyright holder to publish these figures specifically under the CC BY 4.0 license, or (b) remove the figures from your submission Thank you for your suggestion. We have incorporated your feedback and replaced the figure/map. The locations were identified using Google Earth and then pinned for clarity using customizer.com. The revised map has been included in the revised manuscript.

5 Please review your reference list to ensure that it is complete and correct. If you have cited papers that have been retracted, please include the rationale for doing so in the manuscript text, or remove these references and replace them with relevant current references. Any changes to the reference list should be mentioned in the rebuttal letter that accompanies your revised manuscript. If you need to cite a retracted article, indicate the article’s retracted status in the References list and also include a citation and full reference for the retraction notice. Thank you for your valuable feedback. We have carefully reviewed and corrected the references according to your suggestions.

Additional Editor Comments:

1 Authors need to address orphan claims in the discussion sections with fig number or cited reference. The stats applied needs to be explained in terms of significance. Optionally, a more mechanistic insight into the observed phenomenon seems to be lacking in the discussion section. The manuscript needs revision according to the expert reviewers. There are no suggested further/revised experiments, and therefore a minor revision is recommended. We appreciate your valuable feedback. In line with your suggestion, we have inserted the appropriate figure numbers into the relevant sections of the discussion in the revised manuscript. Additionally, we have incorporated a detailed discussion of the observed phenomena across all three formulations, broken down by season, as suggested. Relevant statistics have been applied, and the description of analysis under results section has been provided in the revised manuscript. Grateful for your constructive input, which has helped to enhance the focus of the manuscript.

Review Comments to the Author

1 Reviewer #1: The manuscript needs to be revised to improve it. The title needs to be more concise, the abstract needs to correctly capture the findings and statements of statistical significance of means of populations will be important. There are also certain questions that have to be addressed probably in the discussion section. A more detailed file with comments to the authors is attached. Agreed. Consistent to the suggestions we have revised the manuscript. As suggested we have revised the title to “A large-scale trial in Chennai, India shows effective reduction of Anopheles stephensi populations by the biolarvicide Natular”.

2 Reviewer #2: The study titled “A large-scale field trial in Chennai, India found that Natular®, a biolarvicide, was effective in reducing populations of Anopheles stephensi, a major malaria vector” investigates the evaluation of effectiveness of three newly formulated biolarvicides, Natular 20.6% Emulsifiable Concentrate (EC), 2.5% G30 granular, and 7.48% tablet for direct application (DT), against Anopheles stephensi. The research aims to assess the best formulation and the dosage to make recommendations for better larval management of Anopheles spps. The finding of the study and the recommendation makes the study of high impact to control the diseases like malaria.

The insecticide resistance against the available anti-larval insecticide Temephos calls for urgent action in vector control strategies to combat the vector in India. Insecticide resistance in the diseases vectors poses a significant threat to public health. This study provides an alternative solution using the new formulations with up to 100% larval control. These Natular formulations serve as potent bio larvicides, offering a valuable tool for An. stephensi control within the Integrated Vector Management (IVM) program. Development of new formulations is crucial for developing effective vector control strategies and preventing the spread of malaria parasites.

I think that this is a very nice and important contribution to the existing literature offering an alternative to insecticide resistance in An. stephensi. It should be published once the following issues have been addressed.

The authors don’t discuss the mechanism of action for the three different formulations.

Have any cross-resistance studies been conducted for the formulations. We are grateful for the positive response and appreciate the comments.

The formulations of Natular contain technical-grade Spinosad as the active ingredient. Spinosad is a mixture of two metabolites: spinosyn A (C41H65NO10) and spinosyn D (C42H67NO10), both produced by the soil bacterium Saccharopolyspora spinosa. Its mode of action involves excitation of the insect nervous system, resulting in involuntary muscle contractions, prostration with tremors, and eventual paralysis. Spinosad functions as a nicotinic agonist, altering the activity of both nicotinic and GABA-gated ion channels. This depolarizes insect neurons, leading to neuron excitation. These details are discussed in the Test Materials section under Materials and Methods.

To the best of our knowledge, there are no published reports of cross-resistance for NATULAR formulations (DT 7.48%, EC 20.6%, and GR 2.5%). This may be due to the fact that NATULAR is a relatively new or niche product, and there may not yet be extensive research available on the subject.

Reviewers' comments:

1. Title page

A large-scale trial in Chennai, India found that Natular, a biolarvicide, was effective in reducing populations of Anopheles stephensi, a major malaria vector

Comment: Change the title to “A large-scale trial in Chennai, India shows effective reduction of Anopheles stephensi populations by the biolarvicide Natular” We are grateful for the suggestions. As suggested we have changed the title to

“A large-scale trial in Chennai, India shows effective reduction of Anopheles stephensi populations by the biolarvicide Natular’ in the revised manuscript

2 Study area:

Tamil Nadu, India, under the jurisdiction of the Greater Chennai Corporation (GCC). Chennai is geographically located at 13° 5' 24" N and 80° 16' 12" E along the Coromandel Coast, in the northern region of Tamil Nadu (citation?).

Presently, the estimated population of the city is 7.1 million, distributed among three regions: North Chennai, Central Chennai, and South Chennai. The city

encompasses a total area of 426 square kilometres (citation?). We are grateful to the reviewers for pointing this out. We have incorporated the relevant sources/citations at the appropriate places in the revised manuscript

3 Due to its littoral and equatorial location, the city experiences minimal fluctuations in seasonal temperatures. Minimum temperatures in January range between 18 and 20 degrees Celsius, whereas maximum

temperatures in May and June reach 38 to 42 degrees Celsius (citation?).

The northeast monsoon, which

occurs between mid-October and mid-December, provides the majority of the annual precipitation of about 1400 millimeters (citation?).

Annually, between 53.6% and 78.8% of the state's malaria cases have been reported

in the city alone over the last two decades (citation?).

As of October 2017, 3,303 of the total 4,671 malaria

cases in the state were reported in Chennai. Chennai documented a total of 1,471 malaria cases

in 2018 and 771 malaria cases in 2019, with Plasmodium vivax constituting the prevailing

parasite species, accounting for 95 to 97% of the reported cases (citation?). We are grateful to the reviewers for pointing this out. We have incorporated the relevant sources/citations at the appropriate places in the revised manuscript

4 In 500-liter overhead (sintex) tanks (OHTs), Natular formulations were evaluated against An. stephensi at the optimum field application dosage determined in the phase II trial (citation?). We have incorporated the citation in the revised manuscript.

5 Comment: what was the value of the optimum field application dosage determined in the phase II trial and could you give a citation? The optimal field application dosages for the selected habitats are as follows: 1) 0.274 mg active ingredient (AI)/L for the 20.6% EC formulation, 2) 0.06 mg AI/L for the 2.5% G30 formulation, and 3) 1 tablet per 200 L for the 7.48% DT formulation. The results of the Phase II trial have not yet been published; therefore, they are cited as VCRC, unpublished report.

6 Results

General comment:

The values for reduction of mosquito population by spinosad in the abstract were not consistent with those presented in the results section. The authors should modify the abstract accordingly. The abstract has been revisited and modified accordingly.

7 Results

General comment:

In the text, include statements of the statistical significance of the 3 different test formulations and their respective control populations for the seasons investigated. Give the p values as appropriate. As suggested we have included a section “Statistical Analysis” in the results section, with appropriate p values, in the revised manuscript.

8 Discussion

In a recent publication, Clarke Environmental Technologies (I) Ltd., located in Mumbai, India, introduced three novel spinosad (Natular) formulations:

7.48% DT, 20.6% EC, and 2.5% G30 (citation?). These formulations demonstrated efficacy against An. stephensi larvae in both simulated field (phase II) and laboratory (phase I) conditions (ICMR VCRC, unpublished report).

Comment: Could the authors cite the recent publication from Clarke Environmental Technologies? Thank you for bringing this to our attention. We sincerely apologize for the typographical errors. We have revised the sentence to accurately convey the intended meaning and have cited the company's website for reference: Natular Larvicide.

9 Conclusions:

Furthermore, in line with the Integrated Vector Management program, Natular (spinosad) formulations appear to be effective biolarvicides for managing An. stephensi and other container-breeding mosquitoes, as evidenced by their longer residual activity, efficacy, and favourable toxicological and environmental profile.

Comment: Limit conclusion to An. stephensi since other container-breeding mosquitoes were not investigated in this study. Agreed. As suggested we have limited the conclusion to Anopheles stephensi.

10 References

Comment: Check to make sure that all references comply with the journal referencing style. All the references are modified according to the journal referencing style.

11 General questions: Given that spinosad has been in existing use for several years, what is the significance of this particular study? How will the results from this study translate into real life field applications? While the 3 formulations show varying efficacy in different seasons, the authors do need to recommend to readers which of the 3 formulations is the best and why. Will these 3 formulations be purchased or given free to residents to put in their water tanks, wells and sumps around their surroundings, and what are short or long term health risks of spinosads use in domestic water to humans? Spinosad has been used since 1999 on more than 250 crops and in consumer and animal health uses in over 85 countries. Its first use in public health began in 2009, when it was introduced as the active ingredient in Clarke’s Natular® brand of mosquito larvicides. The active ingredient in Natular products, spinosad, is the only active ingredient currently used for mosquito control that is designated by IRAC (Insect Resistance Action Committee) as Group 5. The benefit of this is that it has no cross-resistance with existing products – making Natular an excellent option for resistance management.

The Natular 7.48% DT formulation is well-suited for An. stephensi larval control at 1 tablet per 200 L with an 8-week application interval.

Spinosad was the first mosquito larvicide active ingredient registered under the EPA’s

Reduced Risk program due to its reduced risk to human health and non-target organisms when compared to other available alternatives.

When applied as indicated on the label for control of mosquito larvae, Natular will not

endanger human or animal health. Spinosad is not toxic to mammals. Spinosad is not

carcinogenic, not genotoxic, and is not a reproductive or developmental toxin.

Prior to registering a product, the EPA evaluates products thoroughly to be sure it can be used safely, within minimum risk to humans, animals and the environment. Spinosad has been approved by the EPA for use in a variety of outdoor aquatic areas that breed mosquitoes, including in residential and recreational areas.

---

## [Decision Letter · Decision Letter 1]

26 Feb 2025

Dear Dr. Shriram,

Thank you for submitting your manuscript to PLOS ONE. After careful consideration, we feel that it has merit but does not fully meet PLOS ONE’s publication criteria as it currently stands. Therefore, we invite you to submit a revised version of the manuscript that addresses the points raised during the review process.

We look forward to receiving your revised manuscript.

Kind regards,

Yash Gupta, Ph.D.

Academic Editor

PLOS ONE

Additional Editor Comments:

Kindly, carefully address the ethical concerns

Reviewers' comments:

Reviewer's Responses to Questions

**Comments to the Author**

Reviewer #3: All comments have been addressed

Reviewer #4: (No Response)

2. Is the manuscript technically sound, and do the data support the conclusions?

Reviewer #3: Yes

Reviewer #4: (No Response)

3. Has the statistical analysis been performed appropriately and rigorously?

Reviewer #3: Yes

Reviewer #4: (No Response)

4. Have the authors made all data underlying the findings in their manuscript fully available?

Reviewer #3: Yes

Reviewer #4: (No Response)

5. Is the manuscript presented in an intelligible fashion and written in standard English?

Reviewer #3: Yes

Reviewer #4: (No Response)

Reviewer #3: The manuscript is excellent and its results are very useful for malaria control programs in countries affected by the disease, including Iran.

It is better to reference the following articles

1. Mosquito surveillance and the first record of morphological and molecular-based identification of invasive species Aedes (Stegomyia) aegypti (Diptera: Culicidae), southern Iran

2. Epidemiology of malaria in Nikshahr, Sistan and Baluchestan province, Southeast Iran, during 2004-2010

Reviewer #4: The study addresses a critical public health issue—malaria control—by evaluating environmentally safer biolarvicides as alternatives to chemical larvicides. This study represents a Phase III evaluation of a Spinosad-based formulation.

Major Comments:

1. Lines 30-32: The statement, “Larval Source Management (LSM) works well in some eco-epidemiological situations because it can target both resting and biting mosquitoes both indoors and outside, which is a key part of lowering residual transmission [1],” is inaccurate. LSM focuses on controlling mosquito immature stages (eggs, larvae, and pupae) in aquatic habitats, rather than directly impacting adult mosquitoes. It does not target resting or biting adults, whether indoors or outdoors. Please revise accordingly.

2. Lines 40-41: The statement, “organophosphate insecticide, temephos, is used to control mosquitoes that spread malaria and arboviruses,” should specify that temephos is used as a larvicide to control mosquito larvae rather than adult mosquitoes.

3. The mode of action of Spinosad should be included in the Intro section rather than in the M&M section to provide proper context for the study.

4. The statement, “Spinosad, acting as a nicotinic agonist, disrupts the functionality of ion channels regulated by GABA and nicotinic adenosine triphosphate (NAP), resulting in the depolarization and subsequent excitation of neurons in insects,” requires correction. To the best of my knowledge, there is no known 'nicotinic adenosine triphosphate (NAP)' receptor in insect neurophysiology. Please verify this term and provide appropriate references. Additionally, Spinosad binds to nicotinic acetylcholine receptors (nAChRs) and acts as an allosteric activator rather than a direct agonist like nicotine or neonicotinoids. It is good to justify all statements with proper references.

5. The Results section presents statistical inferences, but it does not specify the actual statistical test performed. Clearly state which statistical tests were used for analysis to ensure transparency and reproducibility.

6. Explicitly state whether ethical approval was obtained for this study. Additionally, clarify how informed consent was handled, particularly in cases where the product was applied in household overhead tanks.

**Do you want your identity to be public for this peer review?** For information about this choice, including consent withdrawal, please see our Privacy Policy

Reviewer #3: **Yes: ** Hamzeh Alipour

Reviewer #4: **Yes: ** Om P Singh

---

## [Author Response · Author response to Decision Letter 2]

5 Nov 2025

The response to the reviewers has been attached in the revised submission.

---

## [Editor Report · Decision Letter 2]

13 Nov 2025

A large-scale trial in Chennai, India shows effective reduction of Anopheles stephensi populations by  the biolarvicide Natular

PONE-D-24-05784R2

Dear Dr. Shriram,

We’re pleased to inform you that your manuscript has been judged scientifically suitable for publication and will be formally accepted for publication once it meets all outstanding technical requirements.

Kind regards,

Yash Gupta, Ph.D.

Academic Editor

PLOS ONE
---

## [Editor Report · Acceptance letter]

PONE-D-24-05784R2

PLOS ONE

Dear Dr. Shriram,

I'm pleased to inform you that your manuscript has been deemed suitable for publication in PLOS ONE. Congratulations! Your manuscript is now being handed over to our production team.

Kind regards,

on behalf of

Dr. Yash Gupta

Academic Editor

PLOS ONE